# Variational Pseudo Marginal Methods for Jet Reconstruction in Particle Physics

*Hanming Yang [1,*]*                                                    *hy2781@columbia.edu*
*Antonio Khalil Moretti [1,2,*]*                              *antoniomoretti@spelman.edu*
*Sebastian Macaluso [3]*                                      *seb.macaluso@gmail.com*
*Philippe Chlenski [1]*                                              *pac@cs.columbia.edu*
*Christian A. Naesseth [4]*                                      *c.a.naesseth@uva.nl*
*Itsik Pe'er [1]*                                                      *itsik@cs.columbia.edu*

[1] *Department of Computer Science, Columbia University*
[2] *Department of Computer Science, Spelman College*
[3] *Telefonica Research*
[4] *University of Amsterdam*

*Equal contribution*

**Reviewed on OpenReview:** *https://openreview.net/forum?id=pCapRF2vFf*

## Abstract

Reconstructing jets, which provide vital insights into the properties and histories of subatomic particles produced in high-energy collisions, is a main problem in data analyses of collider physics. This intricate task deals with estimating the latent structure of a jet (binary tree) and involves parameters such as particle energy, momentum, and types. While Bayesian methods offer a natural approach for handling uncertainty and leveraging prior knowledge, they face significant challenges due to the super-exponential growth of potential jet topologies as the number of observed particles increases. To address this, we introduce a Combinatorial Sequential Monte Carlo approach for inferring jet latent structures. As a second contribution, we leverage the resulting estimator to develop a variational inference algorithm for parameter learning. Building on this, we introduce a variational family using a pseudo-marginal framework for a fully Bayesian treatment of all variables, unifying the generative model with the inference process. We illustrate our method's effectiveness through experiments using data generated with a collider physics generative model, highlighting superior speed and accuracy across a range of tasks.

## 1 Introduction

Reconstructing jets in particle physics deals with estimating a high-quality hierarchical clustering. A comprehensive approach to this process also involves inference on model parameters, which could provide insights into our understanding of quantum chromodynamics (QCD), i.e. the theory of the strong interaction between quarks mediated by gluons. Hierarchical clustering forms a natural data representation of data generated by a Markov tree, and has been applied in a wide variety of settings such as entity resolution for knowledge-bases (Green et al., 2012; Vashishth et al., 2018), personalization (Zhang et al., 2014), and jet physics (Cacciari et al., 2008; Catani et al., 1993; Dokshitzer et al., 1997; Ellis & Soper, 1993). Typically, work has focused on approximate methods for relatively large datasets (Bateni et al., 2017; Monath et al., 2019; Naumov et al., 2020; Dubey et al., 2014; Hu et al., 2015; Monath et al., 2020; Dubey et al., 2020; Monath et al., 2021). However, there are relevant use cases for hierarchical clustering that require exact or high-quality approximations on small to medium-sized datasets (Greenberg et al., 2020; 2021). This paper deals with one of these use cases: reconstructing the latent hierarchy of *jets* in particle physics. Within this context,

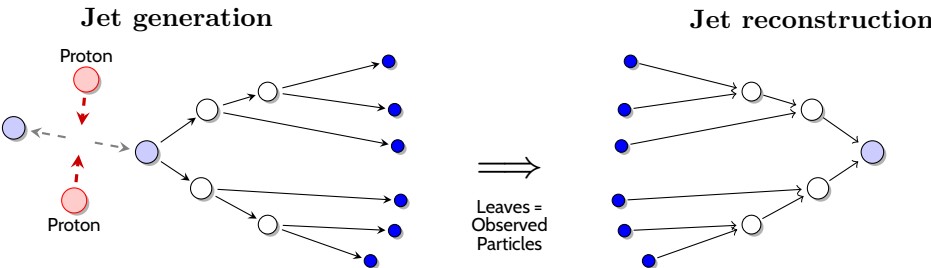

Figure 1: **Jets as binary trees.** Left: Schematic representation of the production of a jet at CERN's LHC. Incoming protons collide, producing two new particles (light blue). Each new particle undergoes a sequence of binary splittings until stable particles (solid blue) are produced and measured by a detector. Right: In jet reconstruction, only the leaf nodes are observed and the tree topology is inferred. Each latent tree topology represents a different possible splitting history.

Bayesian methods provide a natural approach for handling uncertainty, but the super-exponential scaling of the number of hierarchies with the size of the datasets presents significant difficulties, i.e. the number of topologies grows as $(2N-3)!!$, with $N$ being the number of leaves. This super-exponential growth in the space of configurations makes brute force and exact methods intractable.

## 1.1  Jet physics

During high-energy particle collisions, such as those observed at the Large Hadron Collider (LHC) at CERN, collimated sprays of particles called *jets* are produced. The jet constituents are the observed final-state particles that hit the detector and are originated by a *showering process* (described by QCD) where an initial (unstable) particle goes through successive binary splittings. Intermediate (latent) particles can be identified as internal nodes of a hierarchical clustering and the final-state (observed) particles correspond to the leaves in a binary tree. Fig. 1 provides a schematic representation of this process. This results in several possible latent topologies corresponding to a set of leaves. This representation, first suggested in Louppe et al. (2019), connects jets physics with natural language processing (NLP) and biology.

## 1.2  Collider data analysis

A main problem in data analyses of collider physics deals with estimating the latent showering process (hierarchical clustering) of a jet, which is needed for subsequent tasks that aim to identify the creation of different types of sub-atomic particles. The final goal is to perform precision measurements to test the predictions of the Standard Model of Particle Physics and explore potential models of new physics particles, thereby advancing our comprehension of the universe's fundamental constituents. The improved performance of deep learning jet classifiers (Butter et al., 2019) (for the initial state particles) over traditional clustering-based physics observables gives evidence of the limitedness of current clustering algorithms since the *right* clustering should be optimal for classification tasks. Thus, high-quality approximations in this context would be highly beneficial for data analyses in experimental particle physics.

## 1.3  Jet simulators

Currently, there are high-fidelity simulations for jets, such as `Pythia` (Sjostrand et al., 2006), `Herwig` (Bellm et al., 2016), and `Sherpa` (Gleisberg et al., 2009). These simulators are grounded in QCD but make several approximations that introduce parametric modeling choices that are not predicted from the underlying theory, so they are commonly *tuned* to match the data. Many tasks in jet physics can be framed in probabilistic terms (Cranmer et al., 2021). In particular, we consider the challenges of calculating the maximum likelihood hierarchy given a set of leaves (jet constituents), the posterior distribution of hierarchies, as well as estimating the marginal likelihood. Also, these quantities are relevant to tune the parameters of the simulators. While these formulations are helpful conceptually, they are not practical in current high-fidelity simulations

for jets, given that the likelihood is typically intractable (they are implicit models). Thus, we consider `Ginkgo` (Cranmer, Kyle et al., 2021; Cranmer et al., 2019b): a semi-realistic generative model for jets with a tractable joint likelihood and captures essential ingredients of parton shower generators in full physics simulations. In particular, `Ginkgo` was designed to enable implementations of probabilistic programming, differentiable programming, dynamic programming and variational inference.Within the analogy between jets and NLP, `Ginkgo` can be considered as ground-truth parse trees with a known language model.

### 1.4 Related work

#### 1.4.1 Jet clustering

For each jet produced at the LHC, there is an inference task on the latent hierarchy that typically involves 10 to 100 particles (leaves). Though this is a relatively small number of elements, exhaustive solutions are intractable, and current exact methods, e.g., Greenberg et al. (2020; 2021), have limited scalability. The industry standard uses agglomerative clustering techniques, which are greedy and based on heuristics (Cacciari et al., 2008; Catani et al., 1993; Dokshitzer et al., 1997; Ellis & Soper, 1993), typically finding low-quality hierarchical clusterings. Regarding likelihood-based clustering (applied to `Ginkgo` datasets), previous work Greenberg et al. (2020) introduced a classical data structure and dynamic programming algorithm (the *cluster trellis*) that exactly finds the marginal likelihood over the space of configurations and the maximum likelihood hierarchy. Also, an A* search algorithm combined with a *trellis* data structure that finds the exact maximum likelihood hierarchy was introduced in Greenberg et al. (2021). Finally, Cranmer et al. (2022) pairs `Ginkgo` with the cluster trellis (Greenberg et al., 2020), to use the marginal likelihood to directly characterize the discrimination power of the optimal classifier (J. Stuart & Arnold, 1994; Cranmer & Plehn, 2007) as well as to compute the exact maximum likelihood estimate for the simulator's parameters. While these works provide exact algorithms that extend the reach of brute force methods, they have an exponential space and time complexity, becoming intractable for datasets with as few as 15 leaves. For this reason, Greenberg et al. (2020; 2021) also provide approximate solutions at the cost of finding lower-quality hierarchies.

#### 1.4.2 Bayesian inference

A recent body of research has melded variational inference (VI) and sequential search. These connections are realized through the development of a variational family for hidden Markov models, employing Sequential Monte Carlo (Smc) as the marginal likelihood estimator (Maddison et al., 2017; Naesseth et al., 2018; Le et al., 2018; Moretti et al., 2019; 2020; 2021). Within the field of Bayesian phylogenetics (the study of evolutionary histories), various methods have been proposed for inference on tree structures. Common approaches include local search algorithms like random-walk Mcmc (Ronquist et al., 2012) and sequential search algorithms like Combinatorial Sequential Monte Carlo (Csmc) (Bouchard-Côté et al., 2012; Wang et al., 2015). Mcmc methods also handle model learning. Dinh et al. (2017) proposes ppHmc which extends Hamiltonian Monte Carlo to phylogenies. Evaluating the likelihood term in Mcmc acceptance ratios can be challenging. As a workaround, particle Mcmc (Pmcmc) algorithms use Smc to estimate the marginal likelihood and define Mcmc proposals for parameter learning (Wang & Wang, 2020).

Pseudo-marginal methods are a class of statistical techniques used to approximate difficult-to-compute probabilities, typically by introducing auxiliary random variables to form an unbiased estimate of the target probability (Andrieu & Roberts, 2009). Beaumont (2003) introduced a method in genetics to sample genealogies in a fully Bayesian framework. Tran et al. (2016) utilizes pseudo-marginal methods to perform variational Bayesian inference with an intractable likelihood. Our work is a synthesis of Wang et al. (2015) and Moretti et al. (2021) in that we introduce a variational approximation on topologies using Smc and a VI framework to learn parameters.

### 1.5 Contributions of this Paper:

1. We expand upon the Csmc technique introduced by Wang et al. (2015) and the Ncsmc method from Moretti et al. (2021) to introduce a *Combinatorial Sequential Monte Carlo* framework for inferring hierarchical clusterings for jets. The resulting estimators are unbiased and consistent. To the best

of our knowledge, this marks the first adaptation of Smc methods to jet reconstruction in particle physics.

2. We leverage the resulting Smc estimators to develop two approximate posteriors on jet hierarchies and correspondingly two VI methods for parameter learning. We illustrate the effectiveness of both methods through experiments using data generated with `Ginkgo` (Cranmer, Kyle et al., 2021), highlighting superior speed and accuracy across various tasks.

3. In order to circumvent parametric modeling assumptions, we propose a unification of the generative model and the inference process. Building upon the point estimators, we define a distinct variational family over global and local parameters for a fully Bayesian treatment of all variables.

4. We show how partial states and re-sampled indices generated by Smc can be interpreted as auxiliary random variables within a pseudo-marginal framework, thus establishing connections between variational pseudo-marginal methods and Vsmc (Naesseth et al., 2018; Moretti et al., 2021).

## 2 Background

Section 2.1 provides an overview of the `Ginkgo` generative model for jet physics (Cranmer, Kyle et al., 2021). Section 2.3.1 summarizes the approximate inference techniques this work builds upon.

### 2.1 Ginkgo generative model

In this subsection we provide an overview of the generative process in `Ginkgo` as well as jet (binary tree) reconstruction during inference. As mentioned in the introduction, `Ginkgo` is a semi-realistic model designed to simulate a jet. The branching history of a jet is depicted as a binary tree structure $\tau = (\mathcal{V}, \mathcal{E})$ where $\tau$ denotes topology, $\mathcal{V}$ comprises the set of vertices, and $\mathcal{E}$ the set of edges. Each node is characterized by a 4D (energy-momentum) vector $z = (E \in \mathbb{R}^+, \vec{p} \in \mathbb{R}^3)$ where $E$ denotes energy and $\vec{p} = (p_x, p_y, p_z)$ denotes momentum in the respective dimensions. The squared mass $t = t(z) := E^2 - |\vec{p}|^2$ is calculated using the energy-momentum vector $z$. The terminal nodes (or leaf nodes), represented as $\mathbf{X} = \{x_1, \cdots, x_N\}$, correspond to the observed energy-momentum vectors measured at the detector. The tree topology $\tau$ and the energy-momentum vectors associated with internal nodes, denoted as $\mathbf{Z} = \{z_1, \cdots, z_{N-1}\}$, are latent variables in the model.

#### 2.1.1 Generative process

In `Ginkgo` the generative process begins with the splitting of a parent (root) node with invariant mass squared $t_P$ into two children, as shown schematically in Fig. 2 (left). The process is characterized by a cutoff mass squared $t_{cut}$, and a rate parameter $\lambda$ for the exponential distribution governing the decay. During generation, as long as the invariant mass squared of a node exceeds the cutoff value ($t_P > t_{cut}$), that node is promoted to be a parent and the algorithm recursively splits it. The squared masses of each new left (L) and right (R) child nodes ($t_L, t_R$) are obtained from sampling from the exponential distribution $f(t|\lambda, t_P)$ defined in Eq. 1, with parameters specific to each child ($L, R$). Finally, once $t_L$ and $t_R$ are sampled, the corresponding energy-momentum vectors ($z_L, z_R$) for the ($L, R$) nodes are derived from $t_P$, $t_L$ and $t_R$ following energy-momentum conservation rules, i.e. applying a 2-body particle decay (see (Cranmer, Kyle et al., 2021; Cranmer et al., 2019b) for more details). Next, we specify the exponential distribution

$$f(t|\lambda, t_P^i) = \frac{1}{1 - e^{-\lambda}} \frac{\lambda}{t_P^i} e^{-\lambda \frac{t}{t_P^i}} , \tag{1}$$

where the first term $(1 - e^{-\lambda})^{-1}\lambda/t_P^i$ is a normalization factor, $i \in \{L, R\}$, $t_P^L = t_P$, and $t_P^R = (\sqrt{t_P} - \sqrt{t_L})^2$. To satisfy energy-momentum conservation ($\sqrt{t_L} + \sqrt{t_R} < \sqrt{t_P}$), $t_L$ is sampled before $t_R$. Thus, in `Ginkgo` the left child mass squared $t_L$ is sampled first with $t_P^L = t_P$ and then an auxiliary value $t_p^R = (\sqrt{t_P} - \sqrt{t_L})^2$ is calculated to sample the right child mass squared $t_R$ (see Fig. 2 (left)).

There are different types of jets, depending on the type of initial state particle (root of the binary tree). To simulate QCD-like jets, a single $\lambda$ parameter is employed for the entire process. However, for a heavy

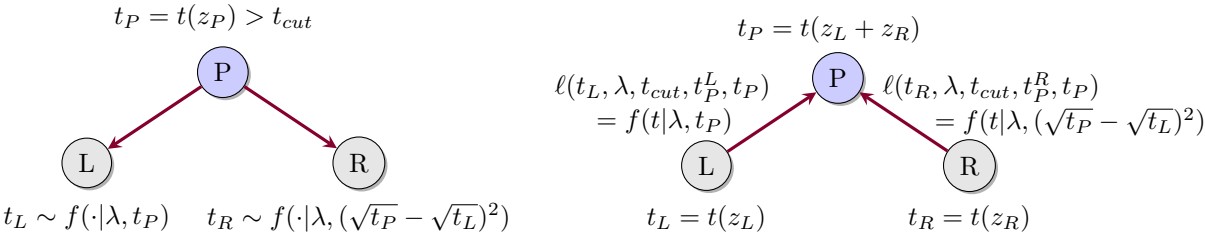

(a) `Ginkgo` generative process.  (b) Node splitting likelihood reconstruction process.

Figure 2: Illustration of the `Ginkgo` generative and reconstruction processes. (a) `Ginkgo` starts with a parent node characterized by a 4-vector $z_p = (E, \vec{p})$ and invariant mass squared $t_P = E^2 - |\vec{p}|^2$. If $t_P$ is greater than the cut off value $t_{cut}$, then the parent node splits (we have a particle decay). The left and right nodes invariant mass squared $(t_L, t_R)$ are sampled from a truncated exponential distribution defined in Eq. 1. (b) The splitting likelihood reconstruction process of a node, defined in Eq. 4 begins with two child nodes $L$ and $R$ along with their respective 4-vectors $z_L$ and $z_R$. The 4-vector for the parent node $P$ is calculated as $z_P = z_L + z_R$ and then $t_P = t(z_P)$. Next, we obtain $t_L = t(z_L)$, $t_R = t(z_R)$ and define $t_P^L = t_P$, and $t_P^R = (\sqrt{t_P} - \sqrt{t_L})^2$. Finally, the left splitting term likelihood $\ell(t_L, \lambda, t_{cut}, t_P^L, t_P)$ and the right one $\ell(t_R, \lambda, t_{cut}, t_P^R, t_P)$ are evaluated.

resonance particle decay, such as a $W$ boson jet, the initial (root node) splitting is governed by a model parameter $\lambda_1$, while the subsequent ones are characterized by $\lambda_2$.

### 2.1.2 Jet reconstruction during inference

In addition to the generative model, we also need to be able to assign a likelihood value to a proposed jet clustering (binary tree) during inference. To do this we use the same general form for the jet's likelihood based on a product of likelihoods over each splitting. In order to evaluate this we need to first reconstruct each parent from its left and right children. Different tree topologies give rise to different $t_P$ values for the inner nodes and thus different likelihoods. Notably, in `Ginkgo` the likelihood of a tree is expressed in terms of the product (in linear space) of all *splitting likelihoods* (specified in Eq. 4 and referred to as the *partial likelihood*) of a parent into two children. The likelihood of a splitting connects parent with child nodes (i.e. we have a likelihood of sampling a child with squared mass $t$ given a parent with squared mass $t_P$). Thus, parent and child nodes are not independent. However, splitting likelihoods of different parent nodes are independent, given a tree. For a set of observed energy-momentum vectors $\mathbf{X} = \{x_1, \cdots, x_N\}$ (leaf nodes), parameters $\theta$, and a tree topology $\tau$, the likelihood of a splitting history can be evaluated efficiently.

### 2.2 Parent node splitting likelihood reconstruction

At inference time, a parent node with energy-momentum vector $z_P$ is obtained by adding its children values $(z_L, z_R)$ as shown schematically in Fig. 2 (right), and $t_P$ is calculated deterministically given $z_P$. The likelihood of a parent splitting into a left (right) child is defined as follows:

$$\ell(t_i, \lambda, t_{cut}, t_P^i, t_P) = \begin{cases} f(t_i|\lambda, t_P^i), & t_P > t_{cut} \\ F_s(t_{cut}, t_P), & t_P \leq t_{cut}, \end{cases} \tag{2}$$

where $i \in \{L, R\}$ (note that to fix a degree of freedom, $t_P^L = t_P$, and $t_P^R = (\sqrt{t_P} - \sqrt{t_L})^2$) and $t_{cut}$ is the cutoff mass squared scale for the binary splitting process to stop (if $t_i \leq t_{cut}$, the corresponding node is a leaf of the binary tree). We introduce the cumulative density function $F_s(t_{cut}, t_P)$ for a given generative process to stop, i.e. the probability of having sampled a value of $t_P < t_{\text{cut}}$, as

$$F_s(t_{cut}, t_P) = \begin{cases} \dfrac{1 - e^{-\lambda t_{cut}/t_P}}{1 - e^{-\lambda}}, & t_P > t_{cut} \\ 1, & t_P \leq t_{cut}, \end{cases} \tag{3}$$

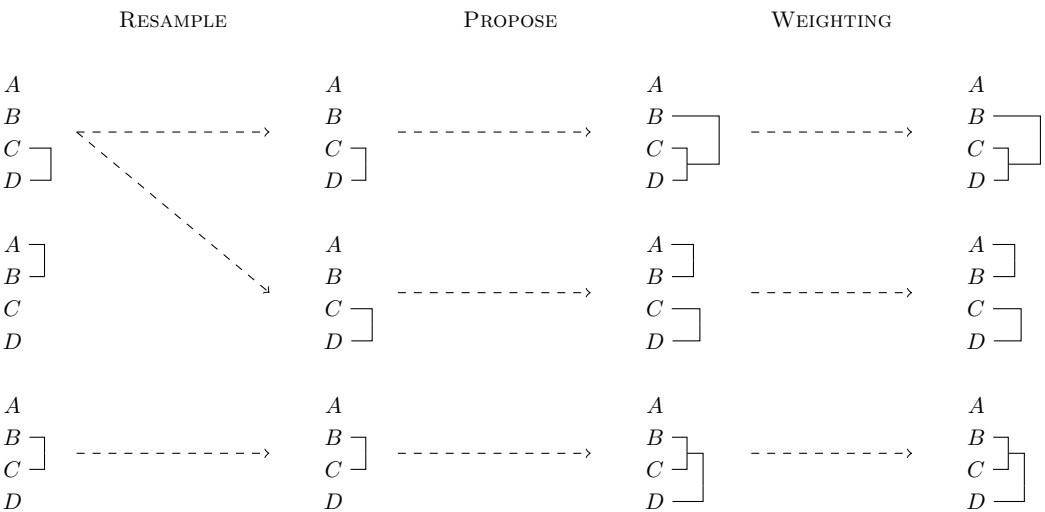

Figure 3: Summary of the CSMC framework: A total of $K$ partial states $\{s_r^k\}_{k=1}^K$ are retained as collections of tree structures encompassing the data set. A partial state is defined as a collection of trees, which start out as singleton particles $A$, $B$, $C$ and $D$. Each iteration within Algorithm 2 comprises three key stages: (1) resampling partial states based on their importance weights $\{w_r^k\}_{k=1}^K$, (2) proposing an expansion of each partial state to form a new one by linking two trees within the forest, and (3) determining the new weights for these new partial states. The illustration above depicts three samples across a jet consisting of observed four particles, denoted as $A, B, C,$ and $D$.

Taking this into account, the probability $\mathcal{F}(t_L, t_R, \lambda, t_{cut}, t_P)$ of a parent node splitting at inference time can be reconstructed as the product of the probability of splitting $(1 - F_s(t_{cut}, t_P))$ times the likelihood of a parent splitting into left and right children as follows:

$$\mathcal{F}(t_L, t_R, \lambda, t_{cut}, t_P) = \frac{1}{4\pi}(1 - F_s(t_{cut}, t_P)) \cdot \ell(t_L, \lambda, t_{cut}, t_P^L) \cdot \ell(t_R, \lambda, t_{cut}, t_P^R). \tag{4}$$

where the factor $\frac{1}{4\pi}$ comes from the likelihood of sampling uniformly over the two-sphere during the 2-body particle decay process. Also, at inference time, given two particles, we assign $t_L \to \max\{t_L, t_R\}$ and $t_R \to \min\{t_L, t_R\}$.

## 2.3 Approximate Inference

### 2.3.1 Bayesian jet reconstruction.

The goal of jet reconstruction is to infer the splitting history and properties of subatomic particles produced during high-energy collisions. This involves estimating various parameters such as particle momenta, positions, and types. Bayesian methods provide a natural framework for modeling uncertainty and incorporating prior information into the reconstruction process. Let $\mathbf{X} = \{x_1, \cdots, x_N\}$ denote the matrix of observed energy-momentum vectors of the Ginkgo model. The posterior distribution can be expressed as follows:

$$P(\tau|\mathbf{X}, \lambda) = \frac{P(\mathbf{X}|\tau, \lambda)P(\tau|\lambda)}{P(\mathbf{X}|\lambda)}. \tag{5}$$

Calculating the denominator requires marginalizing over $(2N-3)!!$ distinct jet topologies which is intractable. The two interconnected tasks for Bayesian jet reconstruction are: (i) computing the normalization constant $P(\mathbf{X}|\lambda)$ by marginalizing all possible candidate topologies:

$$P(\mathbf{X}|\lambda) = \sum_\tau P(\mathbf{X}|\tau, \lambda)P(\tau|\lambda), \tag{6}$$

and (ii): learning or optimizing the likelihood in Eq. 5 obtained by marginalizing Eq. 6: $\hat{\lambda} = \arg\max_{\hat{\lambda} \in \lambda} \log P(\mathbf{X}|\lambda)$. Variational Inference (VI) offers an approach to tackle both tasks for non-trivial posterior distributions.

### 2.3.2 Variational Inference.

VI is a method used to estimate the posterior distribution $P(\tau, \lambda|\mathbf{X})$ when its direct computation is intractable (due to the complexity of marginalizing the latent variables $\tau$). To address this challenge, VI introduces a tractable distribution $Q(\lambda, \tau|\mathbf{X})$ to create a lower bound $\mathcal{L}_{ELBO}$ on the log-likelihood:

$$\log P(\mathbf{X}) \geq \mathcal{L}_{ELBO}(\mathbf{X}) := \mathbb{E}_Q\left[\frac{P(\tau, \lambda, \mathbf{X})}{Q(\tau, \lambda|\mathbf{X})}\right] \tag{7}$$

In the context of Auto-Encoding Variational Bayes (AEVB, both $Q(\tau, \lambda|\mathbf{X})$ and $P(\lambda, \tau, \mathbf{X})$ are jointly trained (Kingma & Welling, 2013; Rezende et al., 2014). To approximate the expectation in Eq.7, Monte Carlo samples from $Q(\tau, \lambda|\mathbf{X})$ are averaged, and these samples are reparameterized using a deterministic function of a random variable that is independent of $\tau$.

Obtaining a feasible approximation for jet structures can be a complex task, leading us to modify and adapt CSMC.

### 2.3.3 Combinatorial Sequential Monte Carlo.

CSMC, tailored for phylogenetic tree models, approximates a sequence of increasing probability spaces, ultimately aligning with Eq. 5 (Wang et al., 2015). CSMC employs sequential importance resampling across $\{r\}_{r=1}^{N-1}$ steps to approximate both the unnormalized target distribution $\pi$ and its normalization constant, denoted as $\|\pi\|$, constituting the numerator and denominator in Eq. 5, by $K$ *partial states* $\{s_r^k\}_{k=1}^K \in \mathcal{S}_r$ to form a distribution (see Wang et al. (2015) or Appendix B),

$$\widehat{\pi}_r = \|\widehat{\pi}_{r-1}\|\frac{1}{K}\sum_{k=1}^K w_r^k \delta_{s_r^k}(s) \qquad \forall s \in \mathcal{S}. \tag{8}$$

CSMC, in contrast to standard SMC techniques, manages a combinatorial set representing the realm of tree topologies alongside the continuous branch lengths—both of which are characteristic features of phylogenies (Wang et al., 2015). Partial states (Monte Carlo samples) are resampled at each rank $r$, ensuring samples remain in high-probability regions, and importance weights are defined as:

$$w_r^k = w(s_{r-1}^{a_{r-1}^k}, s_r^k) = \frac{\pi(s_r^k)}{\pi(s_{r-1}^{a_{r-1}^k})} \cdot \frac{\nu^-(s_{r-1}^{a_{r-1}^k})}{q(s_r^k|s_{r-1}^{a_{r-1}^k})}, \tag{9}$$

where $q(s_r^k|s_{r-1}^{a_{r-1}^k})$ specifies a proposal distribution, $a_{r-1}^k \in \{1, \cdots, K\}$ denote resampled ancestor indices with $\mathbb{P}(a_{r-1}^k = i) = w_{r-1}^i/\sum_{l=1}^K w_{r-1}^l$, and $\nu^-$ is an overcounting correction defined in Wang et al. (2015). Resampled states are then extended via proposal distribution simulations (see Fig. 3). This framework allows for the construction of an unbiased estimate of the marginal likelihood, converging in $L_2$ norm:

$$\widehat{\mathcal{Z}}_{CSMC} := \|\widehat{\pi}_R\| = \prod_{r=1}^R\left(\frac{1}{K}\sum_{k=1}^K w_r^k\right) \to \|\pi\|. \tag{10}$$

The CSMC method is only applicable for sampling toplogies to marginalize over the space of phylogenetic trees, leading us to adapt VCSMC to perform VI.

**Variational Combinatorial Sequential Monte Carlo.** Expanding on the foundation laid by CSMC, Moretti et al. (2021) introduces Variational Combinatorial Sequential Monte Carlo (VCSMC) as an approach

to learn distributions over phylogenetic trees. VCSMC employs CSMC as a means to create an unbiased estimator for the marginal likelihood:

$$\mathcal{L}_{CSMC} \coloneqq \mathbb{E}_Q \left[ \hat{\mathcal{Z}}_{CSMC} \right] . \tag{11}$$

In the same work, Moretti et al. (2021) introduces Nested Combinatorial Sequential Monte Carlo (NCSMC), an efficient proposal distribution, providing an *exact approximation* to the intractable locally optimal proposal for CSMC. We provide a review of NCSMC in Appendix C. The VI algorithms VCSMC and VNCSMC that utilize the estimators $\hat{\mathcal{Z}}_{CSMC}$ and $\hat{\mathcal{Z}}_{NCSMC}$ each introduce a structured approximate posterior that exhibits factorization across rank events. Each state, denoted as $s_r$, is uniquely characterized by its topology, a collection of trees forming a forest, and the corresponding branch lengths. To facilitate reparameterization, discrete terms are either removed from gradient estimates or transformed into Gumbel-Softmax random variables.

## 3 Methods

Section 3.1 adapts the CSMC approach to perform inference on jet tree structures. Section 3.2 reformulates VCSMC for inference on global parameters. Section 3.2.1 utilizes VCSMC methodology to learn parameters as point estimates. Section 3.2.2 defines a prior on the model parameters to construct a variational approximation on both global and local parameters. The resulting approach is interpreted as a variational pseudo-marginal method establishing connections between pseudo-marginal methods (Andrieu & Roberts, 2009) and Variational Combinatorial Sequential Monte Carlo (Naesseth et al., 2018; Moretti et al., 2021).

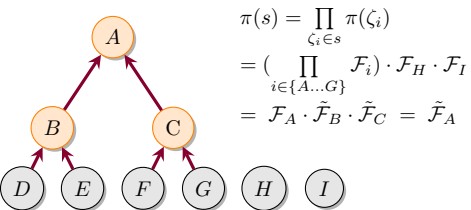

Figure 4: The likelihood $\mathcal{F}_A$ for a sub-tree defined on leaf nodes $D$, $E$, $F$ and $G$ is defined as the recursive product of splitting likelihoods $\mathcal{F}_B$ and $\mathcal{F}_C$. The intermediate target $\pi(s_3)$ for the partial state $s_3$ also includes the probability of singletons $H$ and $I$ denoted $\mathcal{F}_H$ and $\mathcal{F}_I$.

### 3.1 Inference on Tree Structures

Sequential Monte Carlo (SMC) methods (Naesseth et al., 2019; Chopin & Papaspiliopoulos, 2020) are designed to sample from a sequence of probability spaces, where the final iteration converges to the target distribution. However, adapting CSMC for the `Ginkgo` model requires a crucial modification: ensuring that the splitting likelihood at the final coalescent event reflects the dependence on the *entire sub-tree splitting history*, not just the most recent split. This dependence is essential for accurately capturing the recursive structure of the tree. Recall that the splitting likelihood is defined as the product of all *splitting likelihoods* (see Eq. 4).

To achieve this, we reformulate the splitting likelihood to explicitly account for previous splits, as depicted in the recurrence relation:

$$\tilde{\mathcal{F}}(t_L, t_R, \lambda, t_{cut}) = \frac{1}{4\pi} \cdot (1 - F_s(t_{cut}, t_P)) \times \ell(t_L, \lambda, t_{cut}, t_P^L) \cdot \ell(t_R, \lambda, t_{cut}, t_P^R) \tag{12}$$
$$\times \tilde{\mathcal{F}}(t_{LL}, t_{LR}, \lambda, t_{cut}) \cdot \tilde{\mathcal{F}}(t_{RL}, t_{RR}, \lambda, t_{cut}).$$

In the above, the pair $i, j \in (L, R) \times (L, R)$ defines $t_{ij}$ as the mass squared of the $j$ child of the current $i$ coalescent node and $\tilde{\mathcal{F}}(t_L, t_R, \lambda, t_{cut})$ for leaf nodes simply evaluates to 1. Eq. 12 represents the tree (sub-tree) likelihood (with root node having squared mass $t_P$) as the recursive product of splitting likelihoods. Each term brings its normalization, and the overall normalization is correctly expressed as the product of individual ones. In practice we use dynamic programming to maintain a running sum of cumulative log probabilities across rank events. The CSMC resampling step illustrated in Fig. 3 is now dependent upon the full sub-tree splitting history, reflecting the recursive nature of the coalescent process.

In a slight abuse of notation, the probability $\pi(s_r^k)$ of partial state $s_r^k$ (recall $r \in \{1, \cdots, N-1\}$ denotes the coalescent event and $k$ denotes the Monte Carlo sample) is defined as the product of the probabilities of all

disjoint trees $\zeta_i$ in the forest $s$: $\pi(s) = \prod_{\zeta_i \in s} \pi(\zeta_i)$. An illustration of a likelihood for a sub-tree along with the likelihood of a partial state is provided in Fig. 4. The NCSMC algorithm defined in Moretti et al. (2021) can similarly be adjusted to ensure compatibility with this modified framework. The resulting estimators are unbiased and consistent, for proofs see Wang et al. (2015) and Moretti et al. (2021).

## 3.2 Inference on Global Parameters

In Section 3.2.1, we define a variational approximation on topologies, while in Section 3.2.2, we outline fully Bayesian inference using a variational pseudo-marginal framework.

### 3.2.1 Maximum Likelihood

We introduce a variational approximation on $\tau$, using CSMC and the AEVB framework to optimize $\lambda$. Using Eq. 9 and Eq. 10 to define the weights and estimator respectively, along with Eq. 11 and Eq. 12 to evaluate $\pi(s_r^k)$ and form the ELBO, we define a variational family:

$$Q_{\phi,\psi}\left(s_{1:R}^{1:K}, a_{1:R-1}^{1:K}\right) := \prod_{k=1}^{K} q_{\phi,\psi}(s_1^k) \times \prod_{r=2}^{R} \prod_{k=1}^{K} \left[ \frac{w_{r-1}^{a_{r-1}^k}}{\sum_{l=1}^{K} w_{r-1}^l} \cdot q_{\phi,\psi}\left(s_r^k | s_{r-1}^{a_{r-1}^k}\right) \right] . \tag{13}$$

The full factorization of $q_{\phi,\psi}(s_r^k | s_{r-1}^{a_{r-1}^k})$ is written in Eq. 22 and Eq. 23 of the Appendix and depends on the choice of CSMC or NCSMC as an inference algorithm. We utilize our adaptation of CSMC and NCSMC to form the two objectives $\mathcal{L}_{\text{CSMC}}$ and $\mathcal{L}_{\text{NCSMC}}$.

### 3.2.2 Fully Bayesian Inference Using a Variational Pseudo-Marginal Framework

Simulators rooted in quantum chromodynamics are frequently calibrated to align with the data. However, training a simulator separately from the inference process can lead to inefficiencies. To address this, we propose a modification of the posterior distribution defined in Eq. 5 to include a prior on $\lambda$ along with variational parameters to learn the proposal distribution in Eq. 13. We define a log-normal distribution over $\lambda \sim \log\mathcal{N}(\lambda|\mu, \Sigma)$ so that $\lambda$ can be marginalized along with $\tau$. The target distribution can now be specified as follows:

$$P(\tau, \lambda|\mathbf{X}) = \frac{P(\mathbf{X}|\tau, \lambda)P(\tau|\lambda)P(\lambda)}{P(\mathbf{X})} . \tag{14}$$

The generative model parameters can be defined as $\theta := \lambda = \{\mu, \Sigma\}$ or as the output of a neural network and the proposal parameters $\phi = \{\tilde{\mu}, \tilde{\Sigma}\}$ used in the variational approximation to Eq. 14 can be shared or separately trained.

The pseudo-marginal framework is designed to sample from a posterior distribution such as the one defined in Eq. 5 when the marginal likelihood $p(\mathbf{X}|\lambda)$ cannot be evaluated directly. We would normally be interested in computing the posterior distribution over splitting topologies and decay parameters defined in Eq. 5; however, the marginal likelihood $p(\mathbf{X}|\lambda)$ is intractable. Given access to a function $\hat{g}(u; \mathbf{X}, \lambda)$ accepting random numbers $u \sim r(u)$ that can be evaluated pointwise, assume $\hat{g}(u; \mathbf{X}, \lambda)$ returns a non-negative unbiased estimate of $P(\mathbf{X}|\lambda)$:

$$\mathbb{E}_{r(u)}\left[\hat{g}(u; \mathbf{X}, \lambda)\right] = \int \hat{g}(u; \mathbf{X}, \lambda)r(u)du = p(\mathbf{X}|\lambda) . \tag{15}$$

In our setup, $\hat{g}(u; \mathbf{X}, \lambda)$ is defined in Eq. 13 and the auxiliary random variables $u := (s_{1:R}^{1:K}, a_{1:R-1}^{1:K})$ are generated via the CSMC or NCSMC algorithm. Let $p(\lambda, u)$ be a joint target distribution over $\lambda$ and $u$:

$$p(\lambda, u) = \frac{g(u; \mathbf{X}, \lambda)r(u)p(\lambda)}{p(\mathbf{X})} , \tag{16}$$

The integral of the expression above equals one, and its marginal distribution corresponds to the posterior distribution:

$$\pi(\lambda) = \int \frac{g(u; \mathbf{X}, \lambda)r(u)p(\lambda)}{p(\mathbf{X})}du = \frac{p(\mathbf{X}|\lambda)p(\lambda)}{p(\mathbf{X})} . \tag{17}$$

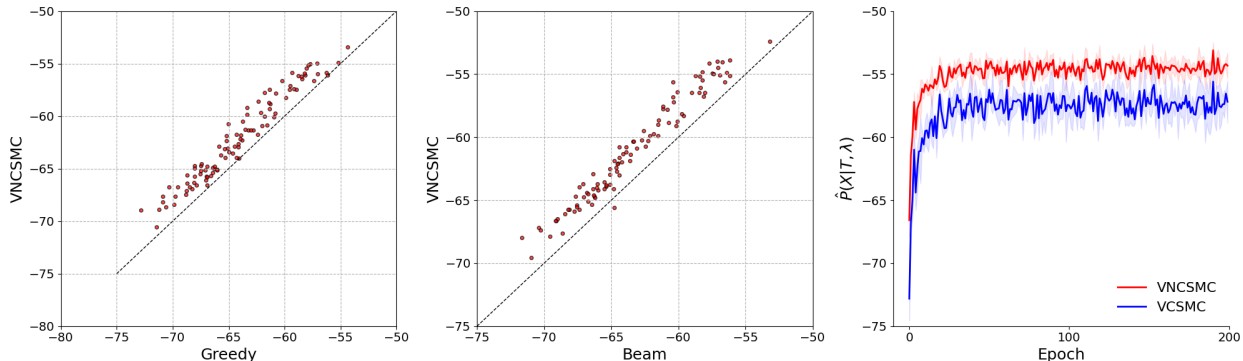

Figure 5: Left: Scatterplot comparing log-conditional likelihood of VNCSMC with $K, M = (256, 1)$ vs Greedy Search and Center: Scatterplot comparing log-conditional likelihood of VNCSMC with $K, M = (256, 1)$ vs Beam Search. Across 100 simulated jets, VNCSMC returns higher likelihood on all 100 cases against Greedy Search and 99 cases against Beam Search. Right: Log-conditional likelihood values for VCSMC (blue) and VNCSMC (red) with $K = \{256\}$ (and $M = 1$) samples averaged across 5 random seeds. VNCSMC achieves convergence in fewer epochs than VCSMC and yields higher values, all while maintaining lower stochastic gradient noise. Additional experiments demonstrating the effect of $K$ appear in Fig. 9 of the Appendix.

This implies that by employing nearly any approximate inference technique to Eq. 16, the resulting marginal distribution of that approximation will serve as an estimate of the actual target distribution. The pseudo-marginal framework by Andrieu & Roberts (2009) designs a Markov Chain $(\theta^i, u^i)$ with Eq. 16 as its target distribution.

Given the conditional likelihood $p(\mathbf{X}|\lambda, \tau)$ we could run MCMC only on the parameter $p(\lambda)$. Instead, we take the approach of sampling $K$ topologies $\tau^k \sim p(\tau|\lambda)$ so that

$$\frac{1}{K} \sum_{k=1}^{K} p(\mathbf{X}|\tau^k, \lambda)\, p(\lambda) \underset{\text{as } K \to \infty}{\Longrightarrow} p(\mathbf{X}|\lambda)\, p(\lambda)\,,$$

where an explicit approximation to $p(\tau|\lambda)$ is defined by the modified CSMC algorithm. This is a form of *variational* pseudo-marginal setup where we are interested in approximately marginalizing out all jet structures. Our distribution estimator which marginalizes $\lambda$ is interpreted analogously.

## 4 Experiments

The industry standard in particle physics uses agglomerative clustering techniques, which are greedy (Cacciari et al., 2012). Beam Search provides a straightforward and significant improvement. Thus, we consider both Greedy and Beam Search as standard, relevant and efficient baselines, also applied in cited works Greenberg et al. (2020; 2021). We simulated 100 jets using `Ginkgo` running comparisons with Greedy Search, Beam Search and Cluster Trellis.

### 4.1 MAP estimate

Fig. 5 (left) provides a scatterplot comparing log-conditional likelihood values. Across 100 simulated jets, VNCSMC with $K, M = (256, 1)$ returns a higher likelihood on all 100 cases against Greedy Search (left) and 99 cases against Beam Search (center). Notably, VNCSMC simultaneously conducts inference and $\lambda$ learning, a feature lacking in Greedy Search and Beam Search, which rely on the user providing $\lambda$ values. Furthermore, VNCSMC yields probability distributions over topologies, while Greedy Search and Beam Search yield single topologies.

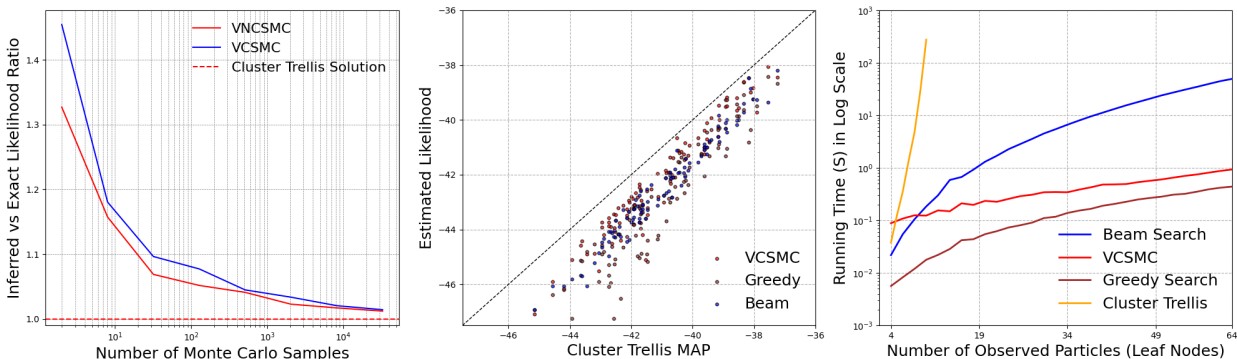

Figure 6: Left: CSMC and NCSMC converge to the exact marginal likelihood as $K$ increases (log scale). Center: Log conditional likelihood returned by VCSMC compared with the exact MAP clustering returned by the cluster trellis Greenberg et al. (2020), Greedy Search and Beam Search, for a dataset of 100 simulated jets generated with `Ginkgo`. Even with $K = 256$ and $N = 12$, VCSMC closely approximates exact cluster trellis values for specific jets. Right: A comparison of the running times, highlighting that VCSMC on $N = 64$ significantly outperforms cluster trellis on $N = 12$. As $N$ increases, the cluster trellis quickly becomes impractical due to its exponential complexity.

Fig. 5 (right) shows the log conditional likelihood $\log \hat{p}(X|\tau, \lambda)$ for VCSMC (blue) and VNCSMC (red) with $K = 256$ (and $M = 1$ for VNCSMC) samples averaged across 5 random seeds. VNCSMC converges in fewer epochs and achieves higher likelihood values with lower gradient noise. Additional experiments illustrating the effect of $K$ appear in Fig. 9 in the Appendix. Larger $K$ yields higher $\log \hat{p}(X|\tau, \lambda)$ values and reduces stochastic gradient noise.

## 4.2 Running Time and Complexity.

We generated jets with $N = \{4, \cdots, 64\}$ leaf nodes and profiled the running time of VCSMC, Cluster Trellis, Greedy Search and Beam Search averaged across 3 random seeds. All experiments were performed on a Google Cloud Platform `n1-standard-4` instance with an Intel Xeon CPU 4 vCPUs and 15 GB RAM without leveraging GPU utilization.

Fig. 6 (left) illustrates convergence of the CSMC and NCSMC conditional likelihood to the exact cluster trellis marginal likelihood as $K$ increases. Fig. 7 plots the inferred Log-Normal Pseudo-Marginal Distribution for the parameters $\mu = (\mu_1, \mu_2)$ of the Heavy Resonance Jet, estimated through NCSMC. Contours of the log-conditional likelihood are shown with stochastic gradient steps taken on $\mathcal{L}_{\text{NCSMC}}$ highlighted in red.

Next, we compare in Fig. 6 (center) the log conditional likelihood returned by VCSMC, Greedy Search and Beam Search with the exact Maximum a Posteriori (MAP) clustering value calculated with the cluster trellis technique described in Greenberg et al. (2020), for a dataset of 100 jets. We see that VCSMC returns high quality hierarchies, with log likelihood values close to the exact ones.

Fig. 6 (right) reports the running times on a log scale in seconds. VCSMC is an order of magnitude faster than Beam Search on $N = 20$ leaf nodes. Beam search entails managing a list of log-likelihood pairs at each level and for each beam size $b$. The given list is sorted, iterated over, and selectively only a single topology is retained at a time. This incurs a complexity of $\mathcal{O}(b^2 N \log b + bN^3 \log N)$. Typically $b > N$, in which case the complexity can be prohibitively slow, but for $b < N^2 \log N$, the complexity becomes $\mathcal{O}(bN^3 \log N)$. In contrast, NCSMC is $\mathcal{O}(KN^3 M)$ and the CSMC is $\mathcal{O}(KNM)$, where $K, N$ and $M$ denote the number of Monte Carlo samples, the number of leaf nodes (observed particles), and the number of subsamples.

## 5 Discussion

In all experiments, $t_{cut}$ is an exogenous variable, and in full physics simulations its value is determined by the energy scale where there is a qualitative change in the theory that describes the process, i.e. we switch from a showering to a hadronization process, described by different models of physics. The difference between the energy of the initial state particle (root of the binary tree) and this energy scale ($t_{cut}$) affects the number of final state particles in the Monte Carlo sampling. Given that Ginkgo does not include a full quantum chromodynamics (QCD) modeling of the splitting likelihood, we choose $t_{cut}$ to control the distribution on the number of final state particles and mimic full physics simulation processes. As such, the exact value is not relevant and the developed algorithms capabilities are independent of it.

Simulators rooted in QCD present significant challenges due to their complex likelihoods. Rewriting these simulators is a substantial endeavor, demanding considerable expertise and effort, with specialists often dedicating entire careers to mastering the intricacies of QCD. Real-world data utilization necessitates strict adherence to the models embedded within these simulators, further complicating the task. Moreover, the lack of currently available powerful quantum computers adds another layer of difficulty, prompting high-energy researchers, such as those at IRIS-HEP, to rely on approximations like Ginkgo (Cranmer et al., 2019a).

The CSMC method (Wang et al., 2015) and the Variational Combinatorial Sequential Monte Carlo (VCSMC) method (Moretti et al., 2021) both use a specific generative model for biology based on a continuous time Markov chain. Our paper represents the first adaptation of Sequential Monte Carlo methods for jet reconstruction in particle physics. This is achieved by adapting the Ginkgo model and redefining the parent node splitting likelihood recursively (Eq. 12) to ensure that, at the final CSMC rank event, the support of the proposal distribution aligns with that of the target distribution.

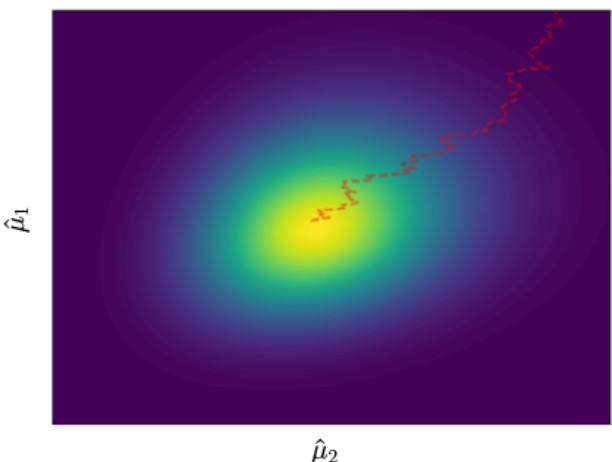

Figure 7: Inferred log-normal pseudo-marginal distribution for the parameters $\hat{\mu} = (\hat{\mu}_1, \hat{\mu}_2)$ of the heavy resonance jet, estimated using NCSMC. Contours represent the log-conditional likelihood, with stochastic gradient steps on $\mathcal{L}_{\text{NCSMC}}$ highlighted in red.

## 5.1 Conclusion

Variational and pseudo-marginal methods are broadly applicable across a range of combinatorial and continuous problems. In Bayesian phylogenetics, these methods facilitate efficient estimation of marginal likelihoods over an equivalent factorial search space of tree topologies, $(2N - 3)!!$, allowing for uncertainty in both tree structure and branch lengths (Moretti et al., 2021; Zhang & Matsen IV, 2019; Zhang et al., 2021). Additionally, in social and biological network analysis, variational techniques are central to hierarchical clustering and stochastic block models for complex, structured latent spaces (Zhou, 2015; Zhang & IV, 2018; Greenberg et al., 2021). Monte Carlo methods, widely used to model self-avoiding random walks on combinatorial structures, provide insight into diffusion and spatial distribution phenomena that bridge into statistical mechanics and more complex particle interaction models (Madras & Slade, 1996; Grosberg et al., 2006). These techniques also extend to probabilistic graphical models and Bayesian networks, where factorized approximations reduce computational burden, supporting inference on high-dimensional data (Zhang & IV, 2018).

We have introduced the first adaptation of CSMC for unbiased and consistent jet reconstruction, proposing approximate posteriors and variational inference (VI) techniques for both point and distribution estimators. Our approach significantly improves both speed and accuracy, paving the way for broader adoption of variational methods in collider data analyses. This work not only provides a robust framework for jet reconstruction but also sets the stage for future advancements in the application of advanced approximate inference methods to high-energy physics experiments.

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

# A   Ginkgo Generative Model

The `Ginkgo` generative process is outlined below: The 2-body decay in the parent rest frame is defined using

---

**Algorithm 1** Toy Parton Shower Generator

---

**Require:** parent momentum $p_p^\mu$, parent mass squared $t_P$, cut-off mass squared $t_{\text{cut}}$, rate for the exponential distribution $\lambda$, binary tree *tree*

 1: **Function** NODEPROCESSING($p_p^\mu$, $t_P$, $t_{\text{cut}}$, $\lambda$, *tree*)

 2: Add parent node to *tree*.

 3: **if** $t_P > t_{\text{cut}}$ **then**

 4:    Sample $t_L$ and $t_R$ from the decaying exponential distribution.

 5:    Sample a unit vector from a uniform distribution over the 2-sphere.

 6:    Compute the 2-body decay of the parent node in the parent rest frame.

 7:    Apply a Lorentz boost to the lab frame to each child.

 8:    **call** NODEPROCESSING($p_p^\mu$, $t_L$, $t_{\text{cut}}$, $\lambda$, *tree*)

 9:    **call** NODEPROCESSING($p_p^\mu$, $t_R$, $t_{\text{cut}}$, $\lambda$, *tree*)

10: **end if**

---

momentum $p_p^\mu = p_L^\mu + p_R^\mu = (\sqrt{s}, 0, 0, 0)$. Due to energy-momentum conservation the child energies are given by

$$E_L = \frac{\sqrt{s}}{2}\left(1 + \frac{t_L}{s} - \frac{t_R}{s}\right) \tag{18}$$

$$E_R = \frac{\sqrt{s}}{2}\left(1 + \frac{t_R}{s} - \frac{t_L}{s}\right) \tag{19}$$

and the magnitude of their 3-momentum is defined

$$|\vec{p}| = \frac{\sqrt{s}}{2}\bar{\beta} = \frac{\sqrt{s}}{2}\sqrt{1 - \frac{2(t_L + t_R)}{s} + \frac{(t_L - t_R)^2}{s^2}} \tag{20}$$

The left and right child momentum are given by $p_L^\mu = (E_L, \vec{p})$ and $p_R^\mu = (E_R, -\vec{p})$ in the parent rest frame. The Lorentz boost $\gamma = \frac{E_p}{\sqrt{t_p}}$ and $\gamma\beta = |\vec{p}_p|/\sqrt{t_p}$. For more information see Cranmer, Kyle et al. (2021); Cranmer et al. (2019b).

# B  Combinatorial Sequential Monte Carlo

We provide an overview of the CSMC algorithm from Wang et al. (2015).

## B.1  Partial States and the Natural Forest Extension

**Definition 1** (Partial State). A rank $r \in \{0, \cdots, N-1\}$ partial state, symbolized as $s_r = (t_i, X_i)$, represents a collection of rooted trees and adheres to the following three conditions:

1. Partial states of different ranks are disjoint, meaning that for any two distinct ranks, $r$ and $s$, there is no overlap between the sets of partial states, written as $\forall r \neq s, \mathcal{S}_r \cap \mathcal{S}_s = \emptyset$.

2. The set of partial states at the smallest rank consists of only one element, denoted as $S_0 = \perp$.

3. The set of partial states at the final rank $R = N - 1$ corresponds to the target space $\mathcal{X}$.

The likelihood, as represented in Eq. 12, and the probability measure $\pi$ are specifically defined within the scope of the target space, denoted as $\mathcal{S}_R = \mathcal{X}$. It's important to note that these definitions apply exclusively to the target space of trees and not the broader sample space encompassing partial states, denoted as $\mathcal{S}_{r<R}$, which consists of forests containing disjoint trees. The Sum-Product algorithm is primarily utilized to derive a maximum likelihood estimate for a tree. However, partial states are explicitly characterized as collections of these disjoint trees or leaf nodes. To extend the target measure $\pi$ to encompass the sample space $\mathcal{S}_{r<R}$, a practical approach is to treat all elements of the jump chain as trees, as elaborated in Wang et al. (2015).

**Definition 2** (Natural Forest Extension). The natural forest extension, denoted as NFE, expands the target measure $\pi$ into forests by forming a product over the trees contained within the forest:

$$\pi(s) := \prod_{(t_i, X_i)} \pi_{Y_i(x_i)}(t_i). \tag{21}$$

One notable advantage of the NFE is its ability to transmit information from non-coalescing elements to the local weight update.

---

**Algorithm 2** Combinatorial Sequential Monte Carlo

**Input:** $\mathbf{Y} = \{Y_1, \cdots, Y_M\} \in \Omega^{N x M}, \theta$

1: Initialization. $\forall k, s_0^k \leftarrow \perp, w_0^k \leftarrow 1/K$.
2: **for** $r = 0$ to $R = N - 1$ **do**
3:  **for** $k = 1$ to $K$ **do**
4:   RESAMPLE

$$\mathbb{P}(a_{r-1}^k = i) = \frac{w_{r-1}^i}{\sum_{l=1}^K w_{r-1}^l}$$

5:   EXTEND PARTIAL STATE

$$s_r^k \sim q(\cdot | s_{r-1}^{a_{r-1}^k})$$

6:   COMPUTE WEIGHTS

$$w_r^k = w(s_{r-1}^{a_{r-1}^k}, s_r^k) = \frac{\pi(s_r^k)}{\pi(s_{r-1}^{a_{r-1}^k})} \cdot \frac{\nu^-(s_{r-1}^{a_{r-1}^k})}{q(s_r^k | s_{r-1}^{a_{r-1}^k})}$$

7:  **end for**
8: **end for**
9: **Output:** $s_R^{1:K}$, $w_{1:R}^{1:K}$

---

## C  Nested Combinatorial Sequential Monte Carlo

We provide a review of the NCSMC algorithm from Moretti et al. (2021). The NCSMC method performs a standard RESAMPLE step (*line 4*), similar to CSMC methods, iterating over rank events. In each iteration, NCSMC explores all possible one-step ahead topologies (($\binom{N-r}{2}$)) and samples *sub-branch* lengths for each of them (*line 5-7*). Importance *sub-weights* or *potential functions* are evaluated for these sampled look-ahead states (*line 8*). The ancestral partial state is then extended to a new partial state by choosing a topology and branch length based on their respective weights (*line 11*). The final weight for each sample is calculated by averaging over the potential functions (*line 12*). For a visual representation of this procedure, please refer to Fig. 8.

---

**Algorithm 3** Nested Combinatorial Sequential Monte Carlo

---

**Input:** $\mathbf{Y} = \{Y_1, \cdots, Y_N\} \in \Omega^{NxM}$, $\theta$

1: Initialization. $\forall k$, $s_0^k \leftarrow \perp$, $w_0^k \leftarrow 1/K$.

2: **for** $r = 1$ **to** $R = N - 1$ **do**

3:     **for** $k = 1$ **to** $K$ **do**

4:        RESAMPLE  $\mathbb{P}(a_{r-1}^k = i) = \frac{w_{r-1}^i}{\sum_{l=1}^K w_{r-1}^l}$

5:        **for** $i = 1$ **to** $L = \binom{N-r}{2}$ **do**

6:            **for** $m = 1$ **to** $M$ **do**

7:              FORM LOOK-AHEAD PARTIAL STATE

$$s_r^{k,m}[i] \sim q(\cdot | s_{r-1}^{a_{r-1}^k})$$

8:              COMPUTE POTENTIALS

$$w_r^{k,m}[i] = \frac{\pi(s_r^{k,m}[i])}{\pi(s_{r-1}^{a_{r-1}^k})} \cdot \frac{\nu^-(s_{r-1}^{a_{r-1}^k})}{q(s_r^{k,m}[i] | s_{r-1}^{a_{r-1}^k})}$$

9:            **end for**

10:        **end for**

11:        EXTEND PARTIAL STATE

$$s_r^k = s_r^{k,J}[I],$$
$$\mathbb{P}(I = i, J = j) = \frac{w_r^{k,j}[i]}{\sum_{l=1}^L \sum_{m=1}^M w_r^{k,m}[i]}$$

12:        COMPUTE WEIGHTS

$$w_r^k = \frac{1}{ML} \sum_{i=1}^L \sum_{m=1}^M w_r^{k,m}[i]$$

13:     **end for**

14: **end for**

**Output:** $s_R^{1:K}$ , $w_{1:R}^{1:K}$

---

## D  Approximate Posteriors

The proposal distribution for our point estimator adaptation of CSMC and the corresponding approximate posterior for VCSMC corresponding to Eq. 5 can be written explicitly as follows:

$$Q_\phi\left(\mathcal{T}_{1:R}^{1:K}, a_{1:R-1}^{1:K}\right) := \left(\prod_{k=1}^K q_\phi(\mathcal{T}_1^k)\right) \cdot \prod_{r=2}^R \prod_{k=1}^K \left[\frac{w_{r-1}^{a_{r-1}^k}}{\sum_{l=1}^K w_{r-1}^l} \cdot q_\phi\left(\mathcal{T}_r^k | \mathcal{T}_{r-1}^{a_{r-1}^k}\right)\right]. \tag{22}$$

ENUMERATE TOPOLOGIES    SUBSAMPLE BRANCH LENGTHS    COMPUTE POTENTIALS

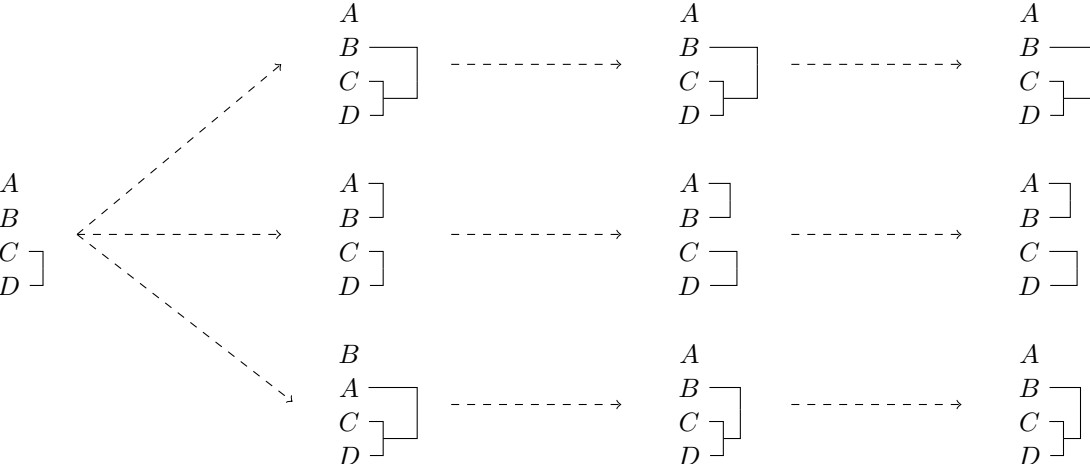

Figure 8: Illustration of the NCSMC framework: In NCSMC, all possible one-step ahead topologies, which amount to $\binom{N-r}{2}$ in total, are systematically enumerated. For the state $A, B, C, D$, the enumerated topologies include (*top*): $A, B, C, D$, (*center*): $A, B, C, D$, and (*bottom*): $B, A, C, D$. Subsequently, $M = 1$ *sub-branch* lengths are stochastically sampled for each edge. Following this, the *sub-weights* or *potentials* are computed (right), and a single candidate is randomly selected proportional to its sub-weight (or potential) to create the new partial state.

In the above, $a_{r-1}^k$ denotes the ancestor index of the resampled random variable and the partial state $s_r^k = \mathcal{T}_r^k$ is sampled by proposing forest $\mathcal{T}_r^k \sim q_\phi(\cdot|\mathcal{T}_{r-1}^{a_{r-1}^k})$ from a UNIFORM distribution. Similarly, the proposal distribution for the global posterior defined in Eq. 14 is expressed where $\lambda_r^k \sim q_\psi(\cdot|\lambda_{r-1}^{a_{r-1}^k}) = \log N(\cdot|\tilde{\mu}, \tilde{\Sigma})$:

$$Q_{\phi,\psi}\left(\mathcal{T}_{1:R}^{1:K}, \Lambda_{1:R}^{1:K}, a_{1:R-1}^{1:K}\right) := \left(\prod_{k=1}^K q_\phi(\mathcal{T}_1^k) \cdot q_\psi(\lambda_1^k)\right) \cdot \prod_{r=2}^R \prod_{k=1}^K \left[\frac{w_{r-1}^{a_{r-1}^k}}{\sum_{l=1}^K w_{r-1}^l} \cdot q_\phi\left(\mathcal{T}_r^k|\mathcal{T}_{r-1}^{a_{r-1}^k}\right) \cdot q_\psi\left(\lambda_r^k|\lambda_{r-1}^{a_{r-1}^k}\right)\right].$$

$$(23)$$

The NCSMC method detailed in Algorithm 3 can also be used to form an unbiased and consistent estimator of the log-marginal likelihood $\widehat{\mathcal{Z}}_{NCSMC}$ and a variational objective which we refer to as $\mathcal{L}_{NCSMC}$:

$$\mathcal{L}_{NCSMC} := \mathbb{E}_Q\left[\log \hat{\mathcal{Z}}_{NCSMC}\right], \qquad \widehat{\mathcal{Z}}_{NCSMC} := \prod_{r=1}^R \left(\frac{1}{K}\sum_{k=1}^K w_r^k\right). \qquad (24)$$

# E  Log Conditional Likelihood $\hat{P}(X|\tau,\lambda)$

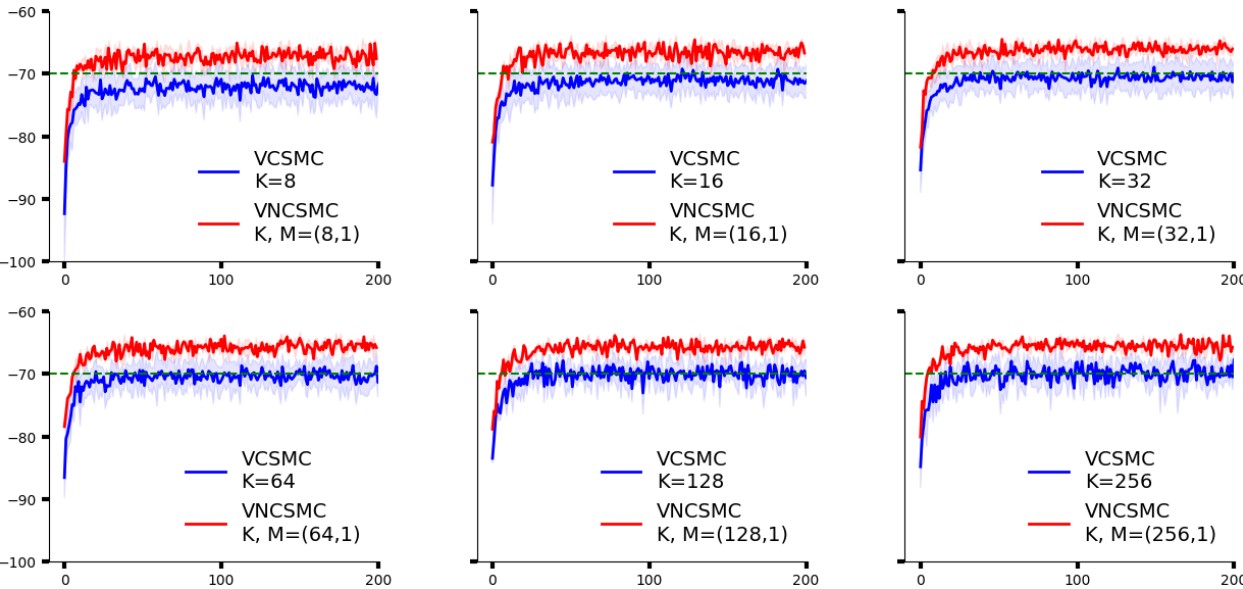

Figure 9: Log-conditional likelihood $\hat{P}(X|\tau,\lambda)$ values for VCSMC (blue) with $K = \{8, 16, 32, 64, 128, 256\}$ samples and VNCSMC (red) with $K = \{8, 16, 32, 64, 128, 256\}$ and $M = 1$ samples averaged across 5 random seeds. Greater values of $K$ result in a more constrained ELBO and higher log-likelihood values while reducing stochastic gradient noise. VNCSMC with $K \geq 8$ explores higher probability spaces than the likelihood returned by the simulator, as depicted by the green trace for reference. VNCSMC achieves convergence in fewer epochs than VCSMC and yields higher values, all while maintaining lower stochastic gradient noise. Notably, even VNCSMC with $(K, M) = (8, 1)$ in the top-left plot (red) outperforms VCSMC with K = 256 in the bottom-right plot (blue).

# F  Implementation Details

## F.1  Invalid Partial States when Coalescing Particles

Physics imposes several constraints on which pairs of particles are impossible to coalesce. We must consider these constraints as we are building trees from leaf to root, coalescing particles (represented as nodes) at every iteration. The following are the conditions

1. $t > 0$ for any node.

2. $t_p > t_{cut}$ for all inner nodes.

3. $t_p > max(t_l, t_r)$ for all inner nodes.

Recall that the ELBO is a function of the weight matrix, which is of dimensions $(R, K)$, and contains all weights of the $K$ particles across $R$ iterations. Each entry in the matrix represents the corresponding weight of some partial state.

In VCSMC and VNCSMC, resampling ensures that we not only extend upon partial states of valid non-zero probability, but we also arrive at $K$ valid trees at the final rank event. We note that both Greedy Search and Beam Search often fail to find any valid trees because they reach a set of partial states where no viable tree can be constructed.

