# OpenReview forum: "Variational Pseudo Marginal Methods for Jet Reconstruction in Particle Physics"
_TMLR — Accepted by TMLR_

### Review · Reviewer_zLT2 · 2024-08-07

**Summary Of Contributions:**

This paper provides a new framework to reconstruct the latent hierarchy of jets in particle physics. To reconstruct the structure of a jets in particle collision is an extremely challenging task with a super-exponential growth of possible jet topologies when the number of observed particles increases. Because of this unfavorable scaling, brute force and exact methods are intractable and doomed to fail for large numbers of observed particles.
In this work, the authors propose to use a Variational Combinatorial Sequential Monte Carlo approach to infer the topologies of jets. Furthermore, they also propose to learn the system parameters and also propose a pseudo-marginal framework to combine inference and generation of possible topologies.

In essence, this work suggests to leverage both advantages of generative models to estimate likelihood of proposed topologies and combine this with efficient Sequential Monte Carlo algorithms to achieve better log-likelihood than the standard approaches, e.g., Greedy, Beam search.

**Audience:**

Yes

**Broader Impact Concerns:**

I could not see any concerns regarding the ethical implications of the submitted work.

**Claims And Evidence:**

Yes

**Requested Changes:**

# **Requested Changes**
- Given the content discussed in paragraph 1.4, I'd recommend to rename this "Related work" or similar.

- There are several open questions which require some clarification. The authors could refer to the following section for a list (perhaps not exhaustive) of such questions.

- While I like the graphics from Figure 2, I'd strongly recommend to change the child particles to R (right) and L (left) (instead of C and B) and the parent particle from A to P (parent). This is just a simple modification but would make the understanding of each formula with the corresponding subscript much more intuitive.

- Below equation (2) on page 5, the authors write "(note that $t^L_P = t_P$ , and $t^R_P = (\sqrt{t_P} - \sqrt{t_L} )^2)$". In this notation, what is, in practical terms,  $t^L_P$ and why does that equal $t_P$? Is this attributing all mass initially from the parent particle to the left one as it is the first one to be sampled in the generation process? More elucidations on this step would be appreciated.

- I find Figure 3 hard to parse. In particular, I cannot straightforwardly related this to the text. In particular I am wondering: what is the 'data set' mentioned in the caption? What are input and output of the algorithm? what are the three different rows? What is the output of this algorithm (right-most part of the figure)? I encourage the other to rethink this figure or make it more self-explanatory.

- Small comment. At the end paragraph 2.3.1 the authors write "complex posterior distributions". While it might be obvious to understand what this means, in high energy physics, it is not unusual to find probability distributions with complex term, i.e., chemichal potential in path integral formulation in Lagrangian formalism. I'd advise to change complex to "non-trivial" or "complicated".

- Below equation (14) the authors introduce the model parameters and the proposal parameters (respectively $\theta$ and $\phi$). However, it is highly non intuitive to see where the dependence of $\phi$ comes into place in equation (14) and moreover, the variational family in equation (13) also depend on some parameters $\psi$ which have not been defined.

- In equation (17) the resulting distribution depends on $\theta$ but there's not such dependence on the rhs of the equality.

- I believe the first paragraph of Section 4.1 is a key point for this work and needs to be highlighted more (perhaps also more in the beginning where contributions are listed)!

- MAP abbreviation does not seem to have been defined.

- I'd recommend to split the 4 plots in Figure 6 into a 2x2 layout to enhance readability. I'd also advice to substantially increase fonts and perhaps use different linestyle in the 2 middle plots to aid the visualization for colorblind people.

- For the left-most plot I'd recommend using different markers to, again, enhance readability.

- I'd recommend to be consistent with the colors in the different plots, e.g., VNCSMC should always have the same color. Same holds for the other methods.

- I find the discussion on beam size $b$ in section 4.2 slightly confusing. As a referee with no prior on the topic, I don't know what a beam size is in this context and as far as I can tell this has not been discussed in the manuscript. Moreover, the equation on the 8th line of the paragraph, namely $b\log < N^2\log N$ seems incomplete as the $\log$ on the left hand side seems to have an argument missing. I'd recommend the author to revise this paragraph.

- I found the rightmost plot in Figure 6 to not add much to the discussion. Of course since the performance has been validated using other metrics one would expect a nice convergence of the loss within SGD, but as far as I can tell this does not seem to add any additional insights. Is my understanding correct or am I overlooking something here? On the other hand, I think it'd be interesting to instead compare CSMC and NSMC with their variational counterpart if any metric is suitable for this task. I perceive that the distinction between such algorithms and their variational counterparts has not been higlighted enough and I believe one additional paragraph discussing this would greatly improve the clarity of the paper.

# **Questions to the Authors**

-  On page 2, paragraph 1.3, the authors write: "While these formulations are helpful conceptually, they are not practical [...] we consider Ginkgo [...]". I am unsure of what the authors mean here. Does that mean they do not consider real data at all, but only data generated by Ginkgo? Or what is the purpose for this claim? If so, why is this the case? Would it be possible to utilize the proposed framework with real world data, e.g., not synthetically generated? Upon carefully reading the manuscript I guess *No* but I'd be great to have some comments from the authors on this.

- On page 2, end of paragraph 1.3, the authors write: "Within the analogy between jets and NLP, Ginkgo can be considered as ground-truth parse trees with a known language model.". I am not quite sure if I fully understand this sentence. I'd appreciate if the authors could elaborate on this.

- The parameter $\lambda$ is introduced in Paragraph 2.1.1. I found it unclear whether this is a learnable parameter or some fixed parameter which is known a priori, e.g., from the physics of the event, type of detector etc. This is important to know in order to understand if that is a variational (learnable) parameter or just a fixed one.

- At the end of paragraph 2.1.1 the authors write "For a set of observed energy-momentum vectors [...] parameters ${\theta}$, and a tree topology $\tau$, the likelihood of a splitting history can be evaluated  efficiently.". What are the parameters $\theta$? Are they model weights? Or are they physical parameters which characterise the physics of the process, e.g., $\lambda, \lambda_1, \lambda_2$? I'd recommend to provide additional explanation around this point in order to avoid any confusion.

- In the Bayesian Inference paragraph, the posterior distribution of a tree topology is expressed as a conditional distribution on the Data X and some parameter $\lambda$. It is not clear to me, as higlighted before, if this is a learnable parameter or if it can be in principle fixed a priori given the physics of a particular event. I believe this is a crucial point of the work and needs more explanation.

- In equation (7) the authors define the ELBO for the problem as a ratio between $P$ and $Q$. To my understanding Q represent the variational sampler conditioned on the data. On the other end, I do not understand what P is. It'd be great if the authors could be more detailed here and explicitly describe each term. Also, the expectation with respect to Q means that parameters $\lambda$ and topoligies $\tau$ are sampled with Monte Carlo? How is this ELBO computed exactly?

- Up to Paragraph 2.3.3 the paper performs a simple review of existing methods (e.g., Ginkgo) and focuses in particular on CSMC and NCSMC and their variational counterpart. I kindly request the authors to correct me in case I have missed any contribution of the manuscript up to page 8.

- From the beginning of Section 3.1, it sounds as if the training is performed by matching CSMC and the Ginkgo output. Yet, it is unclear to the referee what are the parameters optimized in achievieng such a converging criteria.

- Moreover, from section 3.1 I understand that the authors uses the likelihood from the GInkgo model to use the CSMC scheme not only on the previous node but on the full sub-tree spliting history. Is this intuition correct?

- Section 3.2.1: What is AEVB? I think it has not been stated clearly that $\lambda$ is the parameter which needs to be optimized (see previous questions and comments)

- Is $p(X\vert\lambda)$ as defined in equation (15) just a monte carlo estimator of the probability $P(X\vert\lambda$? Why is the latter $P$ capitalized?

- I wonder if the reason why the authors do not compare to CSMC is that there is no straightforward way to obtain a log-likelihood by using such methods.

- Last paragraph os Sec. 4.2 the authors write "Fig. 6 (center right) illustrates convergence of the conditional Csmc and Ncsmc likelihood to the cluster trellis marginal likelihood as K increases.". Judging from the plot this shows convergence of the **variational** CSMC and NCMSC. Am I misunderstanding something here or is there just a typo in the text?

**Strengths And Weaknesses:**

As a disclaimer, the referee is unsure whether some selling points of the paper have been missed. Fort his reason, it'd be greatly appreciated if the authors could highlight any potential misunderstanding in the upcoming review of the manuscript.


# **Strenghts**

- The structure of the paper is well done, and the very detailed introduction makes it accessible to a broad audience. I believe this is an important strength of the paper.
- The manuscript is well-written, and English is appropriate. I could not identify too many typos or problems associated with the writing.
-  The visuals from figure 1 and 2 are particularly useful to gain an intuitive understanding of the problem.
- The numerical experiments clearly show an outperformance of the method discussed in this work compared to established methods.

# **Weaknesses**
- To my understanding, the contributions and novelty of this work sound somehow limited. For instance, in paragraph 1.6, where the paper's contribution are listed, bulletpoint 1 sounds as if an existing algorithm has been applied to the field of Jet reconstruction. I found the other bulletpoints slightly confusing and it was hard for me to disentangle the actual contributions of this work. I believe that reformulating this paragraph by clearly highlighting the novely and contributions of this work would greatly improve the storyline.
- I find the writing hard to follow at times especially in the method (Section 3.1 onwards) part where the reader gets lost in the details and cannot appreciate the actual contribution of this work. Also, some of the math seems wrong or, at least, not clear or consistent with the definitions.
- The problems the authors are trying to solve, and how,  aren't very clear until the very end of the paper.
- Most of the paper focuses on revising previews works, i.e., 8 pages out of 12. While this is good for the broader audience, it makes it hard to asses the actual contribution of the work unless those clearly stand out and clearly emerge from the discussion.

---

> ### Author Response · Authors · 2024-08-15
> **Comments**
>
> Thank you for your careful review of our paper and for the constructive feedback. We have made several updates based on your suggestions and will provide another update shortly. We were uncertain whether additional reviews were forthcoming or if the response window would close soon. Below, we address your comments and requested changes:
>
> * **Related Work:** We have added a dedicated subsection on related work, consolidating subsections 1.4 and 1.5.
> * **Fig. 2 Updates:** We appreciate your suggestion to rename the child particles as L and R in Fig. 2. This change has been implemented for clarity.
> * **Child Invariant Mass Clarification:** The child invariant mass values are indeed constrained by the parent due to energy-momentum conservation. The assignment of $t_L = t_P$ is arbitrary but necessary to fix a degree of freedom. We have clarified this point in the text for greater transparency.
> * **Fig. 3 Explanation:** The three rows in Fig. 3 represent distinct Monte Carlo samples, while the columns depict the three steps within each iteration of CSMC. For further clarity, Algorithm 2 in the appendix which provides detailed pseudocode, outlining the inputs and outputs of CSMC. The "dataset" refers to the four energy-momentum vectors of observed particles A, B, C and D. The algorithm’s input consists of the energy-momentum vectors of each of the M observed particles. We can add further details or clarifications based on your recommendations.
> * **Terminology Update:** To avoid confusion with mathematical terminology, we have replaced the word "complex" with a more precise term.
> * **Parameter Notification Clarification:** We use $\theta$ to denote the generative model or target distribution parameters and $\phi$ for variational approximation or proposal distribution parameters. While sharing parameters between the generative model and the inference network (proposal and target) is possible, we have clarified this in the manuscript.
> * **Eq. 17 Clarification** We have replaced the $\theta$ term in Eq. 17 with $\lambda$ for consistency and added an explanatory note.
>
> We will respond to the remaining questions and comments shortly.

---

> ### Author Response · Authors · 2024-08-18
> **Comments (continued)**
>
> * **Highlight 4.1:** We believe that the most important aspect to emphasize is that the splitting likelihood reflects dependence on sub-tree splits. We have updated and can provide additional changes with specific revisions based on your feedback.
> * **MAP abbreviation**: We have updated the PDF to use "maximum a posteriori" in place of the abbreviation.
> * **Plots:** We will update shortly reflecting the comments on layout, color scheme and markers.
> * **Beam search typo:** Fixed!

---

> ### Author Response · Authors · 2024-08-18
> **Questions: Follow Up**
>
> * **Ginkgo Generative Model:** Regardless of whether real data is used, defining a statistical generative model is essential to specify the splitting likelihood in Eq. 2. We select the Ginkgo model due to its favorable properties for this likelihood, but the same methodology can be adapted to other generative models for jet reconstruction.
>
> * **Jet Reconstruction and Parse Trees:** The Ginkgo model provides a well-defined framework for precise reconstruction of jet structures, analogous to how ground-truth parse trees offer accurate syntactic structures in NLP. Our goal in making this analogy is to underscore that a generative model is vital for understanding and interpreting the underlying structure of the data. We are happy to revise based on your suggestions.
>
> * **Learning \(\lambda\):** In the original CSMC framework, a fixed value for \(\lambda\) is required. However, \(\lambda\) can also be learned by using CSMC to define a variational objective, as outlined in Eq. 11. The expectation within the ELBO is computed using Monte Carlo samples, either over topologies or as detailed in Eq. 13.
>
> * **\(\theta\) Parameters:** \(\theta\) represents the $\lambda_1$ of the root node particle in a QCD-like jet, and, additionally, the $\lambda$ values of subsequent particles in a W boson jet. An alternative parameterization where each particle has its own parameter is also possible.
>
> * **Additional Details on the ELBO:** The term \(P\) denotes the target distribution, which is intractable due to the need to marginalize over \((2N-3)!!\) topologies. In Section 3.1, topologies are sampled using CSMC, whereas in Section 3.2, both topologies and parameters are sampled in a fully Bayesian treatment of all variables.
>
> * **Matching CSMC and Ginkgo Output:** Parameters include the $\lambda_1$ and $\lambda_2$ values required to define QCD or W boson jets.
>
> * **Splitting Tree Likelihood:** Yes, your intuition here is correct.
>
> * **AEVB:** Stands for Autoencoding Variational Bayes. We will update the text to provide additional clarification on this acronym.
>
> * **Comparing to CSMC:** A comparison with CSMC involves hard-coding a value for the trainable parameter \(\lambda\) (or \(\lambda_1\), \(\lambda_2\)). VCSMC reduces to CSMC when a value for trainable parameters is provided.
>
> * **Question about NCSMC Convergence:** We illustrate the convergence of NCSMC to the marginal likelihood value specified by CSMC with fixed parameters.

---

> > ### Comment · Reviewer_zLT2 · 2024-08-19
> > **Acknowledging revision from Authors**
> >
> > Dear authors,
> >
> > Thank you for the thorough revision of your manuscript and for addressing point by point all my requested changes and questions.
> >
> > Would it be possible for you to upload a revised version of the manuscript highlighting the changes such that it'll be easier for me to see all the points addressed in the comments above reflected in the manuscript?
> >
> > Thank you in advance.

---

> > > ### Author Response · Authors · 2024-09-08
> > > **Response to Reviewer Request for Highlighted Revisions**
> > >
> > > Thank you for your suggestion. We have uploaded a revised version of the manuscript with the changes highlighted. We are happy to make further updates based on your feedback as needed.

---

> > > > ### Comment · Reviewer_zLT2 · 2024-09-17
> > > > **Response to Authors**
> > > >
> > > > Dear authors,
> > > >
> > > > Thank you for posting your revision.
> > > >
> > > > I see that the plots still look suboptimal. The plots themselves, as well as axis and tick labels in Figures 5 and 6, are far too small.
> > > > I wonder if you could take care of this as well.
> > > >
> > > > Best wishes

---

> > > > > ### Author Response · Authors · 2024-09-18
> > > > > **Updated Plots**
> > > > >
> > > > > In response to your request, we have split Fig. 6 into two larger figures (now Figs. 6 and 7), each containing two subfigures instead of four, which improves the size of the plots and labels. Unfortunately, due to the page limit, we are unable to further enlarge Fig. 5 within the main text. However, we would like to draw your attention to Fig. 9 in the Appendix, where we provide a larger version of the $\log \hat p(X|\tau,\lambda)$ subfigures from Fig. 5 along with four additional subfigures for various values of K. We are happy to make any additional adjustments based on your feedback, though we are somewhat limited by space constraints.

---

> > > > > > ### Comment · Reviewer_zLT2 · 2024-09-19
> > > > > > **Re: Updated plots**
> > > > > >
> > > > > > Thank you for revising the draft.
> > > > > > The plots in Figs. 6 and 7 are far too large. I recommend making them a little smaller and increasing the labels on the axes andbelieve these are far more important to be visible than having tick labels. I  a very large figure.
> > > > > >
> > > > > > If you resize these appropriately, Figure 5 can also be resized in a 2x2 layout while still fitting the 12-page limit constraint.
> > > > > >
> > > > > > I apologize for being so pedantic, but I think these figures are crucial to the paper as they show the performance of the proposed approach, and therefore, they need to be clear and easy. to read. I'd also recommend to briefly refer to Figure 9 in the caption of Figure 5.
> > > > > >
> > > > > > Thanks for you in advance for your patience in following these advices.

---

> > > > > > > ### Author Response · Authors · 2024-09-19
> > > > > > > **Re: Updated plots**
> > > > > > >
> > > > > > > Thank you for your thorough review of the updated manuscript and for your valuable suggestions. We have revised the plots in Fig. 6 and Fig. 7 to include triplets and have referenced Fig. 9 in the Appendix within the text. Additionally, we removed one of the subfigures and wrapped it in the text to conserve space. We also enlarged the label and tick sizes to enhance readability.

---

> > > > > > > > ### Comment · Reviewer_zLT2 · 2024-09-20
> > > > > > > > **Re: Re: Updated plots**
> > > > > > > >
> > > > > > > > Thank you for following my suggestions! The layout looks much better now.
> > > > > > > > I have the last two small pieces of advice:
> > > > > > > > - In Figure 7 the caption says $\mu=(\mu_1,\mu_2)$ while in the plots $\hat{\mu}_1$ and  $\hat{\mu}_2$ are shown.
> > > > > > > > - Labels and numbers on the axis still need to be enlarged in Figure 6.
> > > > > > > >
> > > > > > > > Other than this, I am happy. Thanks for your work on this.

---

> > > > > > > > > ### Author Response · Authors · 2024-11-04
> > > > > > > > > **Re: Re: Re: Updated plots**
> > > > > > > > >
> > > > > > > > > Thanks for catching this. We have updated the caption and increased the font on Fig. 7.

---

### Review · Reviewer_2M2k · 2024-09-10

**Summary Of Contributions:**

This paper presents a modified combinatorial sequential monte carlo approach for inferring the latent structures of jets. The generative model assumed is based on Ginkgo (from Cranmer et. al.), which models a jet as a binary tree, and splits the a parent node based on a cut off determined by the square of its invariant mass. The standard baseline here are methods based on sequential monte carlo or variational variants. The key insight in this work is to change the way the likelihood is defined in the generative model. Specifically, the likelihood proposed in this work, for a final coalescent event, is forced to depend on the entire sub-tree splitting history (eqn 12). The paper then presents various experiments demonstrating the effectiveness of the proposed modified inference procedures against baselines.

**Audience:**

Yes

**Claims And Evidence:**

Yes

**Requested Changes:**

- It would be great for the authors to discuss whether the modified VNCSMC that they present can be useful beyond the Gingko generative model. It seems like the gains we see here comes from the specific changes that was made to the model's likelihood.

**Strengths And Weaknesses:**

### Strengths
- The paper is well-written and mostly easy to follow.
- I appreciated the discussion in section 2.1, which is essential for those who are not experts in this literature and not familiar with the Ginkgo generative model.

### Weaknesses
- Difficult to separate the contribution here from past work. I am not an expert in this line of work, however, I found it difficult to figure out the pieces that are new in this work and that was already done without going back to read the work of Wang et. al. (2015) and Moretti et. al. (2021). For example, the fact that the estimators here are consistent and unbiased basically defaults to results from both of the previous papers. Overall, it looks like it section 3.2.2 is really the core of the new inference procedure.

- In Figure 6, and 5, I would've like a comparison to a regular CSMC approach, which doesn't take into account the new likelihood, so that we can evaluate the improvements. I assume greedy search (figure 6) is a standard baseline in this lien of work.

---

> ### Author Response · Authors · 2024-09-17
> **Comments**
>
> Thank you for your thoughtful feedback and for noting that our paper is well-written and easy to follow. Regarding the comparison with a regular CSMC approach, this presents certain challenges. Standard SMC methods are designed to align with the target distribution at the final rank event. Without modifying the splitting likelihood to account for the entire sub-tree splitting history, the CSMC likelihood estimate does not align with the target distribution at its final iteration.
>
> To clarify, the splitting likelihood in Eq. 4 represents the probability of a parent node splitting into two children, with the overall likelihood being the product of all such splitting events. In phylogenetics, CSMC employs Felsenstein's Pruning Algorithm to compute the likelihood of a phylogeny by combining the root likelihood with the stationary distribution of a Markov chain. This method allows for the entire tree likelihood to be calculated at the root without revisiting earlier parts of the tree. Consequently, a direct comparison with standard CSMC would not yield a valid likelihood estimate, as the full tree likelihood under Ginkgo is not equivalent to the likelihood estimate designed by CSMC for phylogenetics at the final rank event.
>
> We have revised the manuscript to better articulate this distinction and to clarify the rationale behind our approach.

---

### Review · Reviewer_7rqG · 2024-10-22

**Summary Of Contributions:**

This paper introduces a novel approach to jet reconstruction in particle physics using CSMC. The authors propose a variational inference algorithm for estimating jet structures and learning parameters, integrated with a pseudo-marginal framework for a fully Bayesian approach. They demonstrate the method’s effectiveness through experiments using simulated collider data, showing improvements in speed and accuracy over existing techniques.

**Audience:**

No

**Broader Impact Concerns:**

The work’s broader impact on machine learning, particularly in terms of advancing techniques for Bayesian inference and hierarchical clustering, is not fully explored. The methods presented could be of interest to a wide array of ML fields, but the paper does not emphasize this potential enough. The paper’s impact would be much stronger if it provided explicit connections to other relevant ML areas beyond particle physics.

**Claims And Evidence:**

Yes

**Requested Changes:**

Just several general comments:

1. To appeal to a wider machine learning audience, the authors should explore how the proposed methods could be generalized to other domains. For example, highlighting applications in hierarchical clustering, Bayesian inference, or NLP could make the contribution more impactful for ML researchers.
2. While the authors mention the improvements in speed, more detail on the algorithm’s scalability in practice, especially when applied to real-world collider data, would be valuable.
3. The paper could benefit from additional motivation regarding why ML researchers should care about jet reconstruction in particle physics. How does the advancement in this specialized domain contribute to broader ML challenges?

**Strengths And Weaknesses:**

++:
1. The paper is clear, with a logical flow of ideas and in-depth explanations. The introduction and background sections effectively ground the reader in both particle physics and Bayesian methods.
2. The adaptation of CSMC methods for jet reconstruction in particle physics represents a significant contribution, as these techniques have not been widely applied in this area before. The combination of SMC with variational inference provides a fresh perspective on jet physics problems.
3. The experimental section convincingly demonstrates the efficiency and accuracy of the proposed methods. Results show improvements in speed and precision compared to existing techniques, especially in handling large, complex datasets in collider physics.
4. The unification of the generative model and inference process using the pseudo-marginal framework is particularly appealing for Bayesian approaches to particle physics.

 - -:
1.  While the topic is undoubtedly important for particle physics, it may have limited appeal for a broad machine learning audience. The application is highly specialized, and the novelty of the methodology might not be sufficiently emphasized for researchers focused on more traditional ML domains.
2.  The broader applicability of the proposed methods beyond particle physics is not discussed in depth. It would strengthen the paper if the authors could relate the techniques to other fields, especially in machine learning where hierarchical models and Bayesian inference are widely applicable.

In summary, the paper is well-written, and the contributions are substantial for the field of particle physics. However, to increase its relevance to a broader ML audience, I recommend expanding the discussion of potential applications in other domains, emphasizing the methodological contributions over the specific physics problem, and clarifying the computational complexities.

---

> ### Author Response · Authors · 2024-11-04
> **Comments**
>
> Thank you for your positive feedback on the clarity and contributions of our paper. We have updated the manuscript to include a new subsection, *5.1 Broader Applications in Tree-Structured and Graphical Models*, which outlines the connections between our work and relevant fields within the broader machine learning community.
>
> Specifically, we note that Bayesian phylogenetic inference operates within the same factorial-sized search space of tree topologies, specifically $((2N−3)!!)$. Although our generative models differ, there is substantial literature in ML on variational methods for latent variable phylogenetic models, such as Zhang et al. (2018, 2019, 2021) and Drummond & Rambaut (2007). Additionally, we discuss hierarchical clustering and Bayesian networks as other areas with relevant points of contact.
>
> We also direct your attention to subsection *1.4 Related Work* in the introduction, particularly *1.4.2 Bayesian Inference*, where we discuss connections between variational inference and sequential search using Sequential Monte Carlo as a marginal likelihood estimator (e.g., Maddison et al., 2017; Naesseth et al., 2018; Le et al., 2019; Moretti et al., 2021). This section also reviews key works on pseudo-marginal methods, including Andrieu & Roberts (2009) and Tran et al. (2016), which are particularly relevant to the machine learning community.
>
> We appreciate your insights and believe these updates add value to our manuscript.

---

### Decision · Action_Editor_fqiX · 2024-12-02

**Recommendation:** Accept as is

**Comment:**

For my part, I appologize for the time this paper took in its review process. It was very difficult securing reviews, I think in part because of the topic of the paper.  I want to thank the reviewers and authors for their participation and patience.

**Audience:**

Initially, the worries were to audience.  The paper touches on a more niche area of high energy physics and ML.  Though, with some edits and discussion amongst the reviewers and my own discussion with the authors, the reviewers and I have come to agree that the paper has a home at TMLR and can find an audience there.

**Claims And Evidence:**

The reviewers all unanimously agree that the the paper provides accurate, convincing and clear evidence of its claims.  That doesn't appear to be an issue.